# Mendelian randomization analysis of plasma levels of CD209 and MICB proteins and the risk of varicose veins of lower extremities

Alexandra S. Shadrina[1]◉*, Elizaveta E. Elgaeva[1,2]◉, Ian B. Stanaway[3], Gail P. Jarvik[4], Bahram Namjou[5], Wei-Qi Wei[6], Joe Glessner[7], Hakon Hakonarson[7], Pradeep Suri[8,9,10,11], Yakov A. Tsepilov[1,2]*

1 Laboratory of Recombination and Segregation Analysis, Institute of Cytology and Genetics, Novosibirsk, Russia, 2 Department of Natural Sciences, Novosibirsk State University, Novosibirsk, Russia, 3 Division of Nephrology, Department of Medicine, Kidney Research Institute, University of Washington, Seattle, Washington, United States of America, 4 Department of Medicine (Medical Genetics), University of Washington Medical Center, Seattle, Washington, United States of America, 5 Department of Pediatrics, Cincinnati Children's Hospital Medical Center, Cincinnati, Ohio, United States of America, 6 Department of Biomedical Informatics, Vanderbilt University Medical Center, Nashville, Tennessee, United States of America, 7 Department of Pediatrics, Children's Hospital of Philadelphia, Philadelphia, Pennsylvania, United States of America, 8 Division of Rehabilitation Care Services, VA Puget Sound Health Care System, Seattle, Washington, United States of America, 9 Seattle Epidemiologic Research and Information Center (ERIC), Department of Veterans Affairs Office of Research and Development, Seattle, Washington, United States of America, 10 Department of Rehabilitation Medicine, University of Washington, Seattle, Washington, United States of America, 11 Clinical Learning, Evidence, and Research (CLEAR) Center, University of Washington, Seattle, Washington, United States of America

◉ These authors contributed equally to this work.
* weiner.alexserg@gmail.com (ASS); tsepilov@bionet.nsc.ru (YAT)

**Data Availability Statement:** All relevant data are within the article and its Supporting Information files.

## Abstract

Varicose veins of lower extremities (VVs) are a highly prevalent condition, the pathogenesis of which is still not fully elucidated. Mendelian randomization (MR) can provide useful preliminary information on the traits that are potentially causally related to the disease. The aim of the present study is to replicate the effects of the plasma levels of MHC class I polypeptide-related sequence B (MICB) and cluster of differentiation 209 (CD209) proteins reported in a previous hypothesis-free MR study. We conducted MR analysis using a fixed effects inverse-variance weighted meta-analysis of Wald ratios method. For MICB and CD209, we used data from a large-scale genome-wide association study (GWAS) for plasma protein levels (N = 3,301). For VVs, we used GWAS data obtained in the FinnGen project (N = 128,698), the eMERGE network (phase 3, N = 48,429), and the UK Biobank data available in the Gene ATLAS (N = 452,264). The data used in the study were obtained in individuals of European descent. The results for MICB did not pass criteria for statistical significance and replication. The results for CD209 passed all statistical significance thresholds, indicating that the genetically predicted increase in CD209 level is associated with increased risk of VVs ($\beta_{MR}$ (SE) = 0.07 (0.01), OR (95% CI) = 1.08 (1.05–1.10), $P$-value = 5.9 ×$10^{-11}$ in the meta-analysis of three cohorts). Our findings provide further support that CD209 can potentially be involved in VVs. In future studies, independent validation of our results using data from more powerful GWASs for CD209 measured by different methods would be beneficial.

**Funding:** The work of ASSh, EEE, and YAT was supported by the Russian Ministry of Education and Science under the 5-100 Excellence Programme and by the Ministry of Education and Science of the RF via the Institute of Cytology and Genetics (project #FWNR-2022-0020). Dr. Suri is a Staff Physician at the VA Puget Sound Health Care System. Dr. Stanaway's time was supported by the University of Washington Clinical Learning, Evidence And Research (CLEAR) Center for Musculoskeletal Research Resource Core. The CLEAR Center is a Core Center for Clinical Research (CCCR) funded by the National Institute of Arthritis and Musculoskeletal and Skin Diseases (NIAMS) of the National Institutes of Health under Award Number P30AR072572. The content is solely the responsibility of the authors and does not necessarily represent the official views of the National Institutes of Health. The eMERGE Network phase III was initiated and funded by NHGRI through the following grants: U01HG8657 (Group Health Cooperative/University of Washington); U01HG8685 (Brigham and Women's Hospital); U01HG8672 (Vanderbilt University Medical Center); U01HG8666 (Cincinnati Children's Hospital Medical Center); U01HG6379 (Mayo Clinic); U01HG8679 (Geisinger Clinic); U01HG8680 (Columbia University Health Sciences); U01HG8684 (Children's Hospital of Philadelphia); U01HG8673 (Northwestern University); U01HG8701 (Vanderbilt University Medical Center serving as the Coordinating Center); U01HG8676 (Partners Healthcare/Broad Institute); and U01HG8664 (Baylor College of Medicine). The content of this work is solely the responsibility of the authors and does not necessarily represent the views of the U.S. Department of Veterans Affairs or the United States Government. The funders had no role in study design, data collection and analysis, decision to publish, or preparation of the manuscript.

**Competing interests:** The authors have declared that no competing interests exist.

## Introduction

Varicose veins of lower extremities (VVs) are a very common vascular disease with a prevalence of over 25–30% in many countries [1–4]. Despite intensive research efforts, the precise mechanisms underlying the development of this condition remain unclear [5–7]. According to current understanding, VVs can be defined as a complex disease with multifactorial pathogenesis which may result from the combined action of a number of factors (genetic, lifestyle, hemodynamic, cellular/extracellular, etc.) [5,8]. Pharmacological treatment of VVs is limited to venoactive medications and several other drugs that can reduce the symptoms of chronic venous disease as well as provide a therapeutic benefit for patients with venous leg ulcers [9,10]. However, developing a drug that can prevent VVs formation, recurrence or progression is still challenging.

Mendelian randomization (MR) is a research method to infer potentially causal relationships between phenotypes. This method uses genetic variants associated with "exposure" phenotypes as naturally occurring "randomizations" [11–13]. When applied to molecular traits (e.g., levels of circulating proteins, lipoproteins, metabolites) or other complex phenotypes (e.g., blood pressure) as "exposure" phenotypes and clinical conditions/diseases as "outcome" phenotypes, MR can be used to propose causative factors, to suggest molecular or physiological pathways contributing to disease, and to discover or prioritize potential targets for pharmacological intervention [14,15].

In our recent study, we applied MR to perform a hypothesis-free search for potentially causal relationships between a broad range of phenotypes and VVs [16]. The "exposure" phenotypes included levels of proteins measured by an aptamer-based affinity proteomics platform (SOMAscan) in blood plasma samples of 1,000 individuals [17]. The "outcome" phenotype was the diagnosis of VVs, and the study sample included 408,455 UK Biobank participants [18]. Our study identified two protein traits that passed all the statistical significance thresholds set in our MR and sensitivity analyses. These traits were plasma levels of MHC class I polypeptide-related sequence B (MICB) protein and cluster of differentiation 209 (CD209) antigen (also known as DC-SIGN–dendritic cell-specific intercellular adhesion molecule 3-grabbing non-integrin). As found by MR, a genetically predicted increase in plasma levels of MICB and CD209 was associated with the presence of VVs, so we concluded that these proteins could potentially be involved in VVs pathogenesis [16]. Both MICB and CD209 are involved in the immune system, and their biological roles are well characterized [19–21]. However, an extensive review of the literature did not allow us to formulate any clear hypothesis on how these proteins may influence the development of VVs. Moreover, MICB and CD209 primarily act as cell surface molecules, while our MR analyzes suggested the effects of circulating (secreted or shed) forms. We raised the question of whether our results could be false-positive findings, or whether we found members of yet undiscovered VVs-related pathways. This question can be answered by conducting experimental research. Before undertaking complex and expensive experimental studies, an optimal strategy would be to perform *in silico* replication using independent datasets. Replication reduces the likelihood that the observed effects are chance findings or analysis artifacts. Conversely, when results are not reproduced in independent *in silico* analyzes, this may indicate that further experimental research will not be fruitful. Thus, the aim of the present study was to replicate the MR results for MICB and CD209 levels as an "exposure" and VVs as an "outcome" phenotype in cohorts different from those used in our previous study [16].

## Materials and methods

### Datasets

**SOMAscan data for MICB and CD209 levels.** Genetic association data for plasma levels of MICB and CD209 proteins were obtained from a genetic atlas of the human plasma

proteome (Sun et al. [22]). In that study, relative concentrations of plasma proteins were measured using an expanded version of aptamer-based multiplex protein assay (SOMAscan assay with modified aptamers). The study sample included 3,301 healthy blood donors from the INTERVAL study (recruited in England) genotyped using the Affymetrix Axiom UK Biobank genotyping array [23,24]. Genetic associations were tested by simple linear regression (additive genetic model). Before association testing, relative protein abundances were natural log-transformed and adjusted in a linear regression for age, sex, duration between blood draw and processing and the first three principal components (PCs) of ancestry from multi-dimensional scaling. Then protein residuals were rank-inverse normalized and used as phenotypes in association analysis [22].

**FinnGen data for VVs.** We downloaded genome-wide association study (GWAS) summary statistics for "Varicose veins (I9_VARICVE)" phenotype from the FinnGen research project website (https://finngen.gitbook.io/documentation/v/r3/data-download; data freeze 3). The FinnGen study combines genotype information of samples collected by a network of Finnish biobanks with digital health record data from Finnish national health registries. For VVs phenotype, the case group included 11,006 subjects with International Classification of Diseases, Tenth Revision (ICD-10) code I83 or ICD-8/9 code 454 ("Varicose veins of lower extremities"; 8,554 women and 2,452 men). The control group included 117,692 individuals without codes related to diseases of veins, lymphatic vessels, and lymph nodes (phlebitis and thrombophlebitis; deep vein thrombosis; portal vein thrombosis; other embolism and thrombosis; oesophageal varices; varicose veins (of lower extremities); varicose veins of other sites; other disorders of veins; nonspesific lymphadenitis; other noninfective disorders of lymphatic vessels and lymph nodes; full lists of codes defining each disease endpoint are available on the website: https://www.finngen.fi/en/researchers/clinical-endpoints). Subjects were genotyped with Illumina (Illumina Inc., USA) and Affymetrix arrays (Thermo Fisher Scientific, USA). Genetic associations were tested using a mixed model logistic regression (SAIGE, Scalable and Accurate Implementation of GEneralized mixed model method [25] which accounts for unbalanced case-control ratios and sample relatedness) with the following covariates included in the model: sex, age, 10 PCs, genotyping batch. Further information on data analysis can be found on the FinnGen study website (https://finngen.gitbook.io/documentation/).

**eMERGE data for VVs.** We analyzed data from the Electronic Medical Records and Genomics (eMERGE) network, phase 3 [26]. eMERGE is a network of medical centers in the United States (US) with electronic health record (EHR) data linked to biorepository samples and genomic data [26]. The network was supported by funding from the US National Institutes of Health; eMERGE3 involved nine non-pediatric study sites (Columbia University Health Sciences, Geisinger Health, Partners Healthcare/Harvard University, Kaiser Permanente Washington/University of Washington, Mayo Clinic, Marshfield Clinic, Mt. Sinai Health System, Northwestern University, and Vanderbilt University). Further details regarding genotyping and phenotyping in eMERGE3 have been previously reported [26]. Ancestry was determined by the intersection of self-reported race and principal component analysis (PCA)-based ancestry; analyses were restricted to adults of European ancestry who had at least 1 year of EHR data. Cases and controls were defined using phecodes (https://phewascatalog.org/). Specifically, longitudinal EHR data consisting of ICD-9 and ICD-10 codes were used to identify Phecode 454 ("Varicose veins"). Cases were defined as those with 1 or more instances of Phecode 454 (n = 5,800), and controls had no instances of Phecode 454 (n = 42,629). Genotyping was conducted using Illumina and Affymetrix arrays in 83 batches across the participating sites, with imputation of single nucleotide variants (SNVs) performed using guidelines from the Michigan Imputation Server [27,28] and the Haplotype Reference Consortium (HRC) release 1.1 genome build 37 (hg19) reference panel [29]. Third-degree relatives and

closer relatives were excluded from the analysis to account for interrelatedness. Logistic regression of imputed SNVs with an additive genotype model was conducted in R using the glm() function, adjusting for sex, age, site-specific characteristics, and ancestry PCs 1 to 10. Filters were applied for minor allele frequency (MAF) < 0.005, imputation $r^2 < 0.3$, deviation from Hardy-Weinberg equilibrium (HWE) $P$-value $< 10^{-6}$, genotyping call rate < 0.98, and individual call rate < 0.98.

**UK Biobank data for VVs.** We used genetic association data of UK Biobank study participants available in the Gene ATLAS database (http://geneatlas.roslin.ed.ac.uk/; the second release; data were downloaded in January 2020). Details of the Gene ATLAS study are described in [30]. Details of the UK Biobank study are described in [31–33]. The study cohort was comprised of 452,264 British individuals of European descent. The case group included 12,021 individuals who had ICD-10 code I83 ("Varicose veins of lower extremities") in their medical records, and the control group included 440,243 subjects without this code. Study participants were genotyped with the Affymetrix UK BiLEVE and the Affymetrix UK Biobank Axiom arrays. Associations were tested using a linear mixed model (LMM) method (that allows adjusting for the effect of relatedness), and adjustment was performed for sex, array batch, UK Biobank Assessment Center, age, age$^2$, and the leading 20 genomic PCs as computed by UK Biobank. The polygenic effect was corrected using a leave-one-chromosome-out (LOCO) approach [30,34]. In our study, we converted the linear regression estimates into the approximate logistic regression estimates. Standard errors (SE) were estimated as $SE = 1/\sqrt{varG \times N \times pr \times (1-pr)}$, where $varG$ is the variance of genotype, $N$–sample size, $pr$–VVs prevalence in the UK Biobank cohort. The variance of genotype was estimated under the assumption of Hardy-Weinberg Equilibrium as $varG = 2 \times p \times (1-p)$, where $p$ is allele frequency. $P$-values were converted into $Z$-values, and logistic regression effects ($\beta$) were calculated as $Z/SE$.

## Ethics statement

The human plasma proteome study was approved by the National Research Ethics Service (11/EE/0538) [22]. The UK Biobank study was approved by the North West–Haydock Research Ethics Committee (REC reference: 11/NW/0382) [32]. The FinnGen study was approved by the Ethics Committee of the Helsinki and Uusimaa Hospital District (Nr HUS/990/2017). Human subjects approvals were obtained at each participating site as part of eMERGE3. All participants of these studies completed written informed consent.

## Two-sample Mendelian randomization

Mendelian randomization analysis of potential causal relationships between CD209 and MICB plasma levels and VVs was performed using a fixed effects inverse-variance weighted meta-analysis of Wald ratios (IVW) approach as previously described by the MR-Base collaboration [35]. MR was conducted using the 'MR-Base' R package ('mr()' and 'mr_report()' functions).

Instrumental variables (single nucleotide polymorphisms, SNPs) were selected from the largest available blood plasma proteome GWAS (N = 3,301 individuals) conducted by Sun et al. [22]. We required the selected SNPs (***i***) to be robustly associated with the exposure trait; (***ii***) to be not in linkage disequilibrium (LD) with each other; (***iii***) to have MAF $\geq$ 0.05. The first (***i***) requirement was fulfilled by selecting only those SNPs that are associated with CD209/MICB levels at a genome-wide significance level of $P$-value $< 5 \times 10^{-8}$ in two datasets: in Sun et al. dataset [22] and in the blood plasma proteome GWAS conducted by Suhre et al. (N = 1,000 individuals living in southern Germany) [17]. The latter dataset was used as an "exposure" GWAS in our previous MR study [16]. Besides this, we required the selected SNPs

to have the same direction of effect in both these "exposure" GWASs, so that their association with CD209/MICB can be considered as replicated. The second (**ii**) requirement was met by selecting only one representative SNP per LD region by conducting the iterative LD clumping procedure using PLINK 1.9 software [36,37] (https://www.cog-genomics.org/plink2). Thus, our protocol of instrumental variable (IV) selection involved the following steps: first, we obtained overlapping SNPs between Sun et al. [22] and Suhre et al. [17] datasets; second, we excluded SNPs with MAF < 0.05; third, we selected genome-wide significant and replicated (see above) SNPs; fourth, we performed clumping procedure using the PLINK '—clump' function with a 10,000 kb physical distance, $P$-value $< 5 \times 10^{-8}$ significance threshold, and $r^2 > 0.001$ LD threshold (parameters recommended by the MR-Base, http://app.mrbase.org/). Clumping was performed using Suhre et al. [17] dataset. LD was calculated using 1000 Genomes phase 3 version 5 data for European-ancestry individuals (N = 503). A manual on the clumping procedure can be found at https://zzz.bwh.harvard.edu/plink/clump.shtml. The resulting set of SNPs with association data for CD209/MICB plasma levels is provided in S1 Table.

With these SNPs, we performed three MR analyses: using FinnGen, eMERGE, and Gene ATLAS second release data as an "outcome" GWAS and Sun et al. proteome data [22] as an "exposure" GWAS. Since rs505922 is absent in the FinnGen cohort data, we used a proxy SNP rs576123 for this dataset which is in high LD with rs505922: $r^2 = 0.97$, D' = 0.99 in the FinnGen cohort; $r^2 = 1.00$, D' = 1.00 in Finnish population according to LDlink (https://analysistools.cancer.gov/LDlink/?tab=ldpair; allele rs505922 C is correlated with allele rs576123 C). MR results for FinnGen and eMERGE datasets were meta-analyzed. Finally, we conducted a meta-analysis summarizing all three MR tests. Meta-analysis was performed using a fixed effects IVW approach, and heterogeneity was assessed using the Cochran's Q test. In addition to heterogeneity assessment, we applied a two-sample t-test to compare the MR beta ($\beta_{MR}$) values and their standard errors (SEs) obtained using each pair of datasets. The statistical significance threshold for the Q test and t-tests was set at $P$-value < 0.05.

The criteria for statistically significant and replicated results in the present study were as follows: (1a) $\beta_{MR}$ sign obtained using Gene ATLAS second release data is the same as $\beta_{MR}$ sign obtained in our earlier study [16] (S2 Table); (1b) the $P$-value in this MR analysis is less than $1.1 \times 10^{-5}$ (the threshold used in our previous work [16]); (2a) $\beta_{MR}$ sign in the meta-analysis of MR results obtained using FinnGen and eMERGE data is the same as $\beta_{MR}$ sign obtained in our earlier study [16] (S2 Table); (2b) the $P$-value in this meta-analysis is less than 0.025 (0.05/2) (Fig 1).

## Sensitivity analyses

For CD209, we performed sensitivity tests. First, for each separate IV, we conducted MR analyses for all datasets, meta-analyzed MR results with heterogeneity assessment and then compared $\beta_{MR}$ values and their SEs between each pair of datasets and between a pair of IVs using a two-sample t-test.

Second, we performed additional MR analyses considering IVs suggestively associated ($P$-value $< 5 \times 10^{-7}$) with CD209 level. This was done to increase the number of IVs and perform the tests that could not be performed with a limited number of IVs in the main analysis. For these sensitivity analyses, we used genetic association data for CD209 plasma level from Sun et al. [22] dataset as an "exposure" GWAS and Gene ATLAS second release data for VVs (as the largest dataset for VVs) as an "outcome" GWAS.

IVs were selected using the clumping procedure implemented in PLINK 1.9 software [36,37] with the same settings as described above for the main analysis, except for the statistical

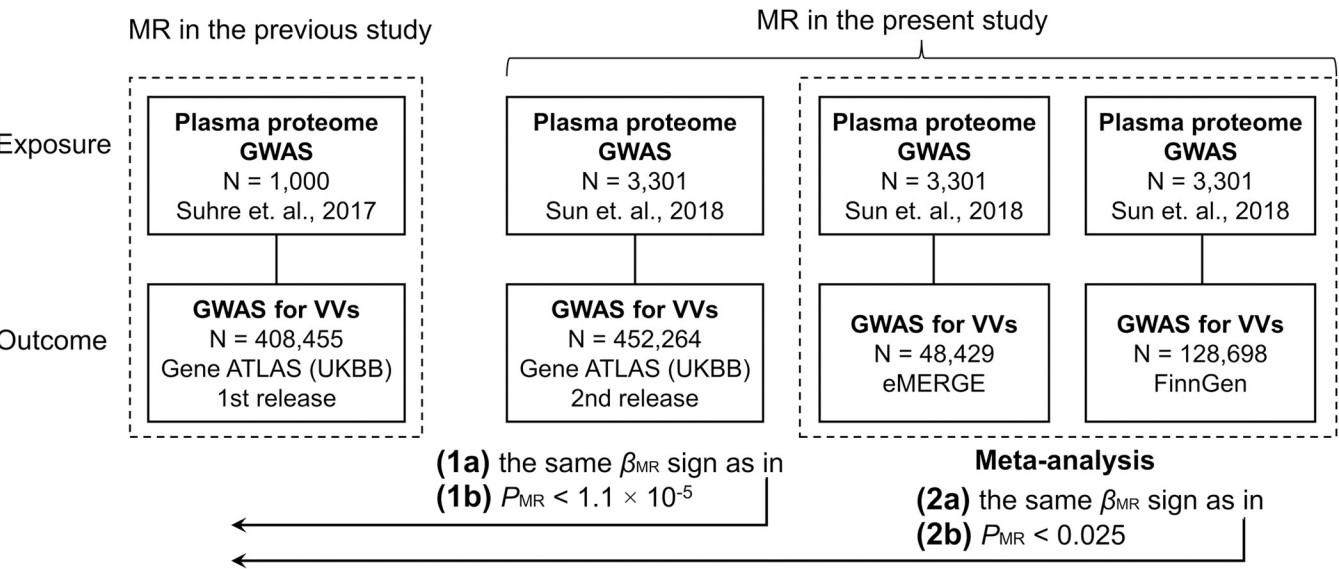

**Fig 1. Criteria for statistically significant and replicated results used in the present study.** GWAS, genome-wide association study; MR, Mendelian randomization; *P*, *P*-value; UKBB, UK Biobank; VVs, varicose veins of lower extremities.

significance threshold of *P*-value = $5 \times 10^{-7}$. Since there were no additional IVs under the relaxed threshold in Suhre at al. [17] GWAS for CD209, we performed clumping using Sun et al. [22] dataset. The list of five selected IVs suggestively associated with CD209 is provided in S3 Table. For rs151212242 variant that is absent in Gene ATLAS data, we used a proxy SNP rs4804224 ($r^2$ = 1.0, D' = 1.0 in European population; allele rs151212242 C is correlated with allele rs4804224 C).

MR analyses were conducted using five MR methods (MR-Egger, Weighted median, IVW, Simple mode, and Weighted mode [35]) integrated into the 'TwoSampleMR' version 0.5.5 R package. Summary statistics for IVs obtained from the exposure and outcome datasets was harmonized using the 'harmonise_data()' function, and MR tests were performed with the 'mr()' and 'mr_report()' functions. Besides the MR tests, the tests for heterogeneity, the test for directional horizontal pleiotropy, and the test identifying the correct direction of effect (Steiger test) embedded into 'TwoSampleMR' package were carried out. The presence of horizontal pleiotropy was additionally assessed using the 'mr_presso()' function of the 'MR-PRESSO' version 1.0 R package [38].

Finally, we used the MR–Robust Adjusted Profile Score (MR-RAPS) approach implemented into the 'mr.raps' version 0.4 R package ('mr.raps.mle.all()' function) to account for the potential bias introduced by selecting weaker IVs [39].

## Results

The results of the IVW MR analysis for MICB and CD209 plasma levels performed using different datasets for the outcome VVs trait are presented in Tables 1 and S4.

### MR analysis for CD209

CD209 passed all the statistical significance thresholds set in our study with the resulting $\beta_{MR}$ (SE) of 0.07 (0.01), odds ratio (OR) of 1.08 with 95% confidence interval (95% CI) of 1.05–1.10, and *P*-value of $5.9 \times 10^{-11}$ in the meta-analysis of MR results for FinnGen, eMERGE, and Gene ATLAS (second release) cohorts. Thus, our results confirmed our previous observations

that the genetically predicted increase in CD209 plasma level is associated with VVs suggesting a causal effect of CD209 on the development of this pathology.

MR analysis for CD209 was conducted using two IVs that met our strict selection criteria (S1 Table; both IVs were the same as those used in our previous study [16]). One of them, rs505922, is located in the gene encoding ABO blood group glycosyltransferases. The variant rs505922 is in LD with rs8176719, a key SNP responsible for blood group O status [40] ($r^2 =$ 0.87, D' = 0.99 in European-ancestry populations according to LDlink). Besides this, rs505922 is in LD with rs507666 ($r^2 = 0.39$, D' = 1.00), which is one of the top SNPs associated with VVs in the "23andMe" GWAS [41] and subsequently replicated in our previous work [42]. According to our estimation, rs505922 explains as much as 40% of the variability in CD209 plasma level [16]. The second IV used in MR was rs8106657. This SNP is located nearly 17 kb from the CD209 gene representing a *cis*-protein quantitative trait locus (pQTL). To assess the impact of each IV on MR results, we repeated a full set of tests with each IV separately (Tables 2 and S5). The results of rs505922-based MR were statistically significant and consistent across all datasets. However, MR with rs8106657 alone did not produce any statistically significant findings, except for a nominally significant result for the Gene ATLAS dataset ($P_{MR} = 0.02$), and $\beta_{MR}$ values were generally less than those obtained in MR with rs505922 alone. For the eMERGE dataset, $\beta_{MR}$ in rs8106657-based MR had a negative sign as opposed to the remaining MR analyses, and for the Gene ATLAS dataset, $\beta_{MR}$ in rs8106657-based MR was 0.10 while for the eMERGE and FinnGen datasets it was close to zero. This prompted us to speculate that the results of the primary MR performed with both IVs could be driven by the effect of rs505922. To check this, we compared $\beta_{MR}$ values (and their SEs) obtained in rs8106657- and rs505922-based MR analyzes assuming that statistically significant differences between these values would indicate a difference in effects. Nevertheless, the two-sample t-tests did not reveal any statistically significant differences between the pairs of $\beta_{MR}$ (and their SEs) for rs8106657 and rs505922 (Tables 2 and S5), neither revealed any differences in $\beta_{MR}$ (and their SEs) between pairs of datasets in the single-IV MR analyses (Tables 2 and S5) and in the primary MR based on both IVs (Tables 1 and S4). Thus, our tests did not provide evidence that rs505922 is fully responsible for the overall effect revealed in our MR analysis.

Further we performed a sensitivity analysis using an extended set of IVs suggestively associated with CD209 plasma level (S3 Table) and five MR methods: MR-Egger, Weighted median, IVW, Simple mode, and Weighted mode. The results were concordant with those obtained in the main IVW MR analysis (S6 Table). Heterogeneity tests provided no evidence for statistically significant heterogeneity in causal effects amongst instruments; horizontal pleiotropy tests (the test based on the MR-Egger regression intercept and the MR-PRESSO global test) showed no evidence for directional horizontal pleiotropy; and the Steiger test did not identify the wrong direction of causality (S6 Table). Finally, we used the MR-RAPS approach to account for the potential bias introduced by weaker IVs. The results of the MR-RAPS analysis were consistent with those of the main analysis (S7 Table) indicating that the detected effect is robust regardless of the number and strength of the selected IVs.

## MR analysis for MICB

For MICB, MR was performed using a single IV rs3094005, which is a *cis-* pQTL located in the *MICB* gene. The results of MR for this protein did not meet the criteria for statistically significant and replicated results set in our study. Firstly, the *P*-value in the MR analysis performed using Gene ATLAS second release data was higher than the threshold used in our previous study based on the first release of the Gene ATLAS database [16] ($3.3 \times 10^{-5}$ vs. $1.1 \times 10^{-5}$). Secondly, the *P*-value in the meta-analysis of MR results for FinnGen and eMERGE cohorts

**Table 1. Results of the IVW Mendelian randomization analysis of the effects of MICB and CD209 plasma levels on VVs.**

| $N_{IV}$[b] | Gene ATLAS (second release)[a] (N = 452,264) | | eMERGE (N = 48,429) | | FinnGen (N = 128,698) | | Meta-analysis eMERGE + FinnGen (N = 177,127) | | Meta-analysis Gene ATLAS[a] + eMERGE + FinnGen (N = 629,391) | | | | Two-sample t-test (comparison of $\beta_{MR}$ and their SEs) | | |
|---|---|---|---|---|---|---|---|---|---|---|---|---|---|---|---|
| | $\beta_{MR}$ (SE) | $P_{MR}$ | $\beta_{MR}$ (SE) | $P_{MR}$ | $\beta_{MR}$ (SE) | $P_{MR}$ | $\beta_{MR}$ (SE) | $P_{MR}$ | $\beta_{MR}$ (SE) | OR (95% CI) | $P_{MR}$ | $P_{het}$[c] | $P_{GE}$[d] | $P_{FE}$[e] | $P_{GF}$[f] |
| **CD209 vs. VVs** | | | | | | | | | | | | | | | |
| 2 | 0.08 (0.02) | 2.0e-7 | 0.06 (0.03) | 0.04 | 0.07 (0.02) | 5.7e-4 | 0.07 (0.02) | 6.1e-5 | 0.07 (0.01) | 1.08 (1.05–1.10) | 5.9e-11 | 0.80 | 0.50 | 0.68 | 0.80 |
| **MICB vs. VVs** | | | | | | | | | | | | | | | |
| 1 | 0.14 (0.03) | 3.3e-5 | -0.09 (0.10) | 0.38 | 0.12 (0.04) | 5.4e-3 | 0.09 (0.04) | 0.03 | 0.12 (0.03) | 1.13 (1.07–1.18) | 4.7e-6 | 0.09 | 0.03 | 0.05 | 0.75 |

Sources of data for the outcome trait (VVs) are indicated in the first raw.

More precise numerical data (with larger number of decimal places) and all OR and 95% CI values are provided in S4 Table.

CI, confidence interval; IV, instrumental variable; MR, Mendelian randomization; OR, odds ratio; P, P-value; SE, standard error.

[a] UK Biobank data available in the Gene ATLAS database (second release) [30].

[b] Number of instrumental variables used in the analysis (listed in S1 Table).

[c] P-value for heterogeneity assessed using Cochran's Q test.

[d] P-value for comparison of $\beta_{MR}$ values and their SEs between the Gene ATLAS and eMERGE datasets using a two-sample t-test.

[e] P-value for comparison of $\beta_{MR}$ values and their SEs between FinnGen and eMERGE datasets using a two-sample t-test.

[f] P-value for comparison of $\beta_{MR}$ values and their SEs between the Gene ATLAS and FinnGen datasets using a two-sample t-test.

was 0.03, which is higher that the Bonferroni-corrected threshold of 0.025 (Tables 1 and S4). Besides this, $\beta_{MR}$ obtained using the eMERGE dataset had the opposite sign to $\beta_{MR}$ obtained in the Gene ATLAS- and FinnGen-based analyses (-0.09 vs. 0.14 and 0.12, respectively). Hence, although the P-value in the meta-analysis of the data for the three cohorts was $4.7 \times 10^{-6}$, we do not consider the association between MICB plasma level and VVs as replicated in our study.

**Table 2. Results of IVW Mendelian randomization analysis of the effect of CD209 plasma levels on VVs considering each selected instrumental variable separately.**

| IV[b] | Gene ATLAS (second release)[a] (N = 452,264) | | eMERGE (N = 48,429) | | FinnGen (N = 128,698) | | Meta-analysis eMERGE + FinnGen (N = 177,127) | | Meta-analysis Gene ATLAS[a] + eMERGE + FinnGen (N = 629,391) | | | | Two-sample t-test (comparison of $\beta_{MR}$ and their SEs) | | |
|---|---|---|---|---|---|---|---|---|---|---|---|---|---|---|---|
| | $\beta_{MR}$ (SE) | $P_{MR}$ | $\beta_{MR}$ (SE) | $P_{MR}$ | $\beta_{MR}$ (SE) | $P_{MR}$ | $\beta_{MR}$ (SE) | $P_{MR}$ | $\beta_{MR}$ (SE) | OR (95% CI) | $P_{MR}$ | $P_{het}$[c] | $P_{GE}$[d] | $P_{FE}$[e] | $P_{GF}$[f] |
| rs505922[g] | 0.08 (0.02) | 2.6e-6 | 0.07 (0.03) | 0.01 | 0.08 (0.02) | 4.2e-5 | 0.08 (0.02) | 1.5e-6 | 0.08 (0.01) | 1.08 (1.06–1.10) | 1.7e-11 | 0.94 | 0.80 | 0.72 | 0.88 |
| rs8106657 | 0.10 (0.04) | 0.02 | -0.01 (0.07) | 0.85 | 0.02 (0.05) | 0.76 | 0.01 (0.04) | 0.89 | 0.05 (0.03) | 1.05 (0.99–1.11) | 0.09 | 0.30 | 0.18 | 0.73 | 0.24 |
| $P_{iv}$[h] t-test | 0.68 | | 0.27 | | 0.25 | | 0.11 | | 0.40 | | | | | | |

Sources of data for the outcome trait (VVs) are indicated in the first raw.

Extended data including all OR and 95% CI values are provided in S5 Table.

CI, confidence interval; IV, instrumental variable; MR, Mendelian randomization; OR, odds ratio; P, P-value; SE, standard error.

[a] UK Biobank data available in the Gene ATLAS database (second release) [30].

[b] Instrumental variables used in the analysis (more details are provided in S1 Table).

[c] P-value for heterogeneity assessed using Cochran's Q test.

[d] P-value for comparison of $\beta_{MR}$ values and their SEs between the Gene ATLAS and eMERGE datasets using a two-sample t-test.

[e] P-value for comparison of $\beta_{MR}$ values and their SEs between FinnGen and eMERGE datasets using a two-sample t-test.

[f] P-value for comparison of $\beta_{MR}$ values and their SEs between the Gene ATLAS and FinnGen datasets using a two-sample t-test.

[g] rs576123 was used as a proxy for rs505922 in the MR analysis for the FinnGen cohort ($r^2 = 0.97$, D' = 0.99).

[h] P-value for comparison of $\beta_{MR}$ values and their SEs between IVs using a two-sample t-test.

## Discussion

In the present study, we used data from three large-scale GWAS for VVs and the largest available GWAS for human plasma proteome to investigate the relationship between plasma levels of MICB and CD209 proteins and VVs via Mendelian randomization. Our results confirmed the association of the genetically predicted increase in CD209 level with VVs revealed in our previous study [16]. The results for MICB did not pass the pre-defined statistical significance criteria (Fig 1), indicating that our previous observation was a false positive.

CD209 (DC-SIGN) is a C-type lectin transmembrane receptor protein primarily expressed by dendritic cells (DC). CD209 acts as a cell adhesion molecule and plays an important role in DC functioning, including interaction with endothelial cells, T-cells, and neutrophils. Besides this, CD209 mediates recognition of a wide variety of pathogens (viruses, bacteria, fungi, parasites) and is involved in their capture and internalization [20,21]. Plazolles et al. [43] demonstrated the presence of a full-length soluble secreted form of CD209 (sDC-SIGN) in several human body fluids such as serum, joint fluids, and bronchoalveolar lavages and showed that its expression appears to be up-regulated upon inflammation. The functional role of the soluble CD209 form and the mechanism of its secretion remain largely unknown, although recent studies have linked changes in its serum level to non-Hodgkin lymphoma [44] and colon cancer [45] and proposed that sDC-SIGN could enhance cytomegalovirus infection [43]. In our MR analyses (both in this and in the previous study [16]), we used data for CD209 determined in plasma using the SOMAscan assay which measures both extracellular and intracellular proteins (including soluble domains of membrane-associated proteins) with a bias towards secreted proteins [22]. Thus, the association with VVs is likely to be shown for the soluble rather than the transmembrane form of CD209. Given the lack of knowledge about the sDC-SIGN function, it is currently challenging to propose a mechanism for its involvement in VVs. However, since the inflammatory response is activated in VVs and is considered part of their pathogenesis [5–8,46,47], we can speculate that this potential link is related to inflammation.

Albeit the association between CD209 and VVs was found in our two studies using independent datasets, the putative causative effect of plasma CD209 level on VVs development has yet to be confirmed in future research. First of all, positive MR results by themselves are insufficient for making a causal claim [48], so *in vitro* and *in vivo* studies are necessary to draw a final conclusion. Besides this, our study has several limitations that must be acknowledged.

The first limitation is a small number of instrumental variables used in the analysis. For CD209, available plasma proteome GWASs [17,22] provide only two genome-wide significant SNPs not in LD with each other (one *cis* and one *trans* pQTL) that can be used as strong IVs in MR. One of them, rs505922, is located in the *ABO* gene in LD with blood group O- and $A_1$-tagging SNPs. These SNPs exert pleiotropic effects on different human traits [49], including the levels of soluble leukocyte adhesion molecules ICAM-1, E-selectin, and P-selectin [50–52]. Since the presence of CD209-independent effect of rs505922 on the risk of VVs can be hypothesized [42], violation of the 'no horizontal pleiotropy' assumption in MR cannot be ruled out. The inclusion of a larger number of IVs in the MR analysis, on the one hand, would enable performing sensitivity tests, and on the other hand, in theory, can lead to a "dilution" of independent pleiotropic effects relative to associations with the trait of interest [53]. Thus, if more powerful plasma proteome GWASs reveal more CD209-associated SNPs in future, replication of our results would be highly beneficial. To test the general reliability of our findings, we compared $\beta_{MR}$ values (and their SEs) obtained in single-IV MR analyzes for both SNPs to check whether MR results could be fully driven by rs505922. However, our tests did not confirm this assumption. Next, we performed a sensitivity analysis using five IVs including weaker ones

associated with CD209 at a suggestive level of statistical significance. This allowed us to use more MR methods and perform additional tests. The results of all MR methods and approaches used were concordant with the results of the main analysis, and no evidence was observed for directional horizontal pleiotropy, heterogeneity between IVs, and the wrong direction of causality.

The second limitation is related to the same method used to measure CD209 in both Suhre et al. [17] and Sun et al. [22] plasma proteome GWASs, the first of which was used in our previous hypothesis-free study [16], and the second of which was used in the present study. If we assume that the aptamer-based method provides biased estimates of CD209 plasma levels, our replication study will suffer from the same problems as the primary study.

The third limitation is that our study only included data obtained from European-ancestry individuals. Thus, the results of our study could not be generalizable to other populations.

Fourth, a phenotyping approach based on the extraction of ICD codes from medical records used in the "outcome" GWASs cannot guarantee that all VVs cases have a confirmed diagnosis and all controls do not have VVs. Of note, the prevalence of VVs was nearly 12% in the eMERGE cohort, 8.6% in the FinnGen cohort, and only 2.7% in the UK Biobank cohort, that is much lower than VVs prevalence estimates obtained in many countries of the Western world (generally over 20%) [1–3]. Nevertheless, this limitation may be compensated for by the large size of the analyzed datasets.

The fifth limitation is that for sensitivity analyses considering IVs suggestively associated with CD209 plasma levels, the same dataset was used for both IV selection and MR analyses, which could lead to so-called "selection bias" [39].

Finally, a confounding effect of inflammation can be proposed if the same inflammatory pathways promote the release of soluble CD209 (sDC-SIGN) [43] and affect the pathogenesis of VVs. Association of IVs with such inflammatory factors, if any, can lead to bias in MR analyses [35].

## Conclusions

Our study provided further evidence that a genetically predicted increase in plasma CD209 level is associated with the risk of VVs, supporting CD209 as a candidate for future studies of the molecular mechanisms of VVs pathogenesis. An independent *in silico* validation of our MR results using an expanded set of instrumental variables from the GWASs with different methods of CD209 measurement (as they become available) would be beneficial.

## Supporting information

**S1 Table. Summary statistics for instrumental variables genome-wide significantly associated with CD209/MICB levels in human blood plasma proteome GWASs.**
(XLSX)

**S2 Table. Results of hypothesis-free 2SMR analysis for CD209 and MICB vs.** VVs obtained in our previous study.
(XLSX)

**S3 Table. Summary statistics for instrumental variables suggestively associated with CD209 level.**
(XLSX)

**S4 Table. Results of the IVW Mendelian randomization analysis of the effects of MICB and CD209 plasma levels on VVs.**
(XLSX)

**S5 Table. Results of the IVW Mendelian randomization analysis of the effect of CD209 plasma levels on VVs considering each selected instrumental variable separately.** (XLSX)

**S6 Table. Results of the Mendelian randomization analysis of the effect of CD209 plasma level on VVs considering instrumental variables suggestively associated with CD209 plasma level.** (XLSX)

**S7 Table. Results of MR-RAPS analysis of the effect of CD209 plasma levels on VVs considering instrumental variables suggestively associated with CD209 plasma level.** (XLSX)

## Acknowledgments

We want to acknowledge the participants and investigators of FinnGen study (https://www.finngen.fi/en), the Gene ATLAS project (http://geneatlas.roslin.ed.ac.uk/), the human plasma proteome study (Sun B.B., et al., 2018; *Genomic atlas of the human plasma proteome*), and the eMERGE Network (https://emerge-network.org/) and thank them for providing genetic data that we have used in our study. We gratefully thank UK Biobank (http://www.ukbiobank.ac.uk/) for establishing a powerful resource for genetic and epidemiological studies.

## Author Contributions

**Conceptualization:** Alexandra S. Shadrina, Yakov A. Tsepilov.

**Data curation:** Elizaveta E. Elgaeva, Yakov A. Tsepilov.

**Formal analysis:** Elizaveta E. Elgaeva, Ian B. Stanaway, Yakov A. Tsepilov.

**Funding acquisition:** Yakov A. Tsepilov.

**Investigation:** Alexandra S. Shadrina, Elizaveta E. Elgaeva, Yakov A. Tsepilov.

**Methodology:** Alexandra S. Shadrina, Elizaveta E. Elgaeva, Yakov A. Tsepilov.

**Resources:** Gail P. Jarvik, Bahram Namjou, Wei-Qi Wei, Joe Glessner, Hakon Hakonarson, Pradeep Suri.

**Supervision:** Yakov A. Tsepilov.

**Writing – original draft:** Alexandra S. Shadrina.

**Writing – review & editing:** Elizaveta E. Elgaeva, Ian B. Stanaway, Gail P. Jarvik, Bahram Namjou, Wei-Qi Wei, Joe Glessner, Hakon Hakonarson, Pradeep Suri, Yakov A. Tsepilov.

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
