## [Decision Letter · Decision Letter 0]

16 Mar 2022

PONE-D-22-02162Mendelian randomization analysis of plasma levels of CD209 and MICB proteins and the risk of varicose veins of lower extremitiesPLOS ONE

Dear Dr. Shadrina,

Thank you for submitting your manuscript to PLOS ONE. After careful consideration, we feel that it has merit but does not fully meet PLOS ONE’s publication criteria as it currently stands. Therefore, we invite you to submit a revised version of the manuscript that addresses the points raised during the review process.

We look forward to receiving your revised manuscript.

Kind regards,

Renato Polimanti, Ph.D.

Academic Editor

PLOS ONE

Journal Requirements:

Additional Editor Comments:

Reviewer #1 highlighted several major issues in the analytic design of the study. I strongly recommend to the authors to fully address Reviewer's 1 requests.

Reviewers' comments:

Reviewer's Responses to Questions

**Comments to the Author**

1. Is the manuscript technically sound, and do the data support the conclusions?

Reviewer #1: Yes

2. Has the statistical analysis been performed appropriately and rigorously? 

Reviewer #1: Yes

3. Have the authors made all data underlying the findings in their manuscript fully available?

Reviewer #1: Yes

4. Is the manuscript presented in an intelligible fashion and written in standard English?

Reviewer #1: Yes

5. Review Comments to the Author

Reviewer #1: Shadrina et al. performed a Mendelian Randomization (MR) study to replicate their previous findings on the effects of the plasma levels of MICB and CD209 proteins on varicose veins of lower extremities (VVs). I consider this study to be sound, and the topic is highly relevant to elucidate the etiology of VVs. I have the following recommendations to improve the manuscript:

Major issues

This study was limited by the small number of instrumental variables used in the analysis (two SNPs). Thus, I suggest performing a sensitivity analysis with suggestive variants. Authors can use the MR–Robust Adjusted Profile Score (MR-RAPS) approach to account for the possible biases introduced by using weaker instrumental variables (IVs).

Please, evaluate the presence of horizontal pleiotropy in the IVs as this could bias the MR estimates.

Please consider evaluating other MR methods (MR-Egger, weighted median, simple mode, and weighted mode) to compare the concordance of the IVW estimates with those obtained with the other MR methods.

Minor issues

Abstract

Please, indicate the meaning of MICB and CD209

Please include the confidence intervals of the effect estimates

Please indicate that the study was performed only in European descents

Introduction

I consider the Introduction to be very long. It includes details that are not necessary to understand the present study, for example, the release of the data and the ICD-code used to define VV in their previous MR. Please, consider a more brief description of this previous study. Also, the extensive description and benefits of MR could be more suitable for the Discussion section.

Methods

Line 156. Please, consider changing the term White by European descents, as the latter term better describes the sample's ancestral origins.

Line 162. I suggest using PCs instead of principal components to be consistent with the abbreviation of PC previously used in this manuscript.

Did the authors evaluate whether the SNPs associated with the exposure were associated with the outcome?

Results

Please, include the confidence intervals for all the reported estimates.

Please report the F-statistic and include the proportion of variance explained (R2) of the IVs.

Discussion

It would be interesting if the authors discuss if there is any potential confounders in the causal association between CD209 and VVs.

Please include as a potential limitation that analyzed data were obtained only from European descents and results could not be generalizable to other populations.

6. PLOS authors have the option to publish the peer review history of their article (what does this mean?). If published, this will include your full peer review and any attached files.

Reviewer #1: No

---

## [Author Response · Author response to Decision Letter 0]

30 Apr 2022

RESPONSE TO COMMENTS

Additional Editor Comments:

Reviewer #1 highlighted several major issues in the analytic design of the study. I strongly recommend to the authors to fully address Reviewer's 1 requests.

Response: Thank you. We performed additional analyses and corrected our manuscript to address Reviewer's 1 requests and improve our study.

Reviewers' comments:

Reviewer #1: Shadrina et al. performed a Mendelian Randomization (MR) study to replicate their previous findings on the effects of the plasma levels of MICB and CD209 proteins on varicose veins of lower extremities (VVs). I consider this study to be sound, and the topic is highly relevant to elucidate the etiology of VVs. 

Response: Thank you. We were very pleased to learn your positive opinion about our study.

Reviewer #1:

I have the following recommendations to improve the manuscript:

Major issues

This study was limited by the small number of instrumental variables used in the analysis (two SNPs). Thus, I suggest performing a sensitivity analysis with suggestive variants. Authors can use the MR–Robust Adjusted Profile Score (MR-RAPS) approach to account for the possible biases introduced by using weaker instrumental variables (IVs).

Response: Thank you for this important suggestion. We performed a sensitivity analysis with an extended set of IVs associated with CD209 level at a suggestive level of statistical significance (5 IVs). The increased number of IVs allowed us to conduct other MR methods as well as to perform the heterogeneity tests, horizontal pleiotropy tests, and the test for the correct direction of effect. We also used the MR-RAPS approach to account for the possible biases introduced by selecting weaker IVs.

We updated the manuscript text with a description of the methods used (added the subsection “Sensitivity analyses” to the Materials and Methods section), added three Supplementary tables (S3 Table, S6 Table, and S7 Table) with the results of sensitivity analyses and GWAS summary statistics for IVs and summarized new results in the Results and Discussion sections.

Reviewer #1:

Please, evaluate the presence of horizontal pleiotropy in the IVs as this could bias the MR estimates.

Response: Thank you. We performed two horizontal pleiotropy tests: the test based on the MR-Egger regression intercept and the MR-PRESSO global test. Both tests did not provide evidence for the presence of directional horizontal pleiotropy. The results are provided in S6 Table.

Reviewer #1:

Please consider evaluating other MR methods (MR-Egger, weighted median, simple mode, and weighted mode) to compare the concordance of the IVW estimates with those obtained with the other MR methods.

Response: Thank you. We conducted sensitivity analyses using MR-Egger, weighted median, simple mode, weighted mode as well as IVW methods. The results were concordant with each other and with those obtained in the main IVW MR analysis using two strong IVs. All results of sensitivity analyses are now presented in S6 Table and described in the text.

Reviewer #1:

Minor issues

Abstract

Please, indicate the meaning of MICB and CD209

Response: Thank you. We added the meaning of MICB and CD209 in the Abstract.

Reviewer #1:

Abstract

Please include the confidence intervals of the effect estimates

Response: Thank you. We included OR and 95% CI in the Abstract.

Reviewer #1:

Abstract

Please indicate that the study was performed only in European descents

Response: Thank you. We added this information in the Abstract.

Reviewer #1:

Introduction

I consider the Introduction to be very long. It includes details that are not necessary to understand the present study, for example, the release of the data and the ICD-code used to define VV in their previous MR. Please, consider a more brief description of this previous study. Also, the extensive description and benefits of MR could be more suitable for the Discussion section.

Response: Thank you. We shortened the Introduction according to these recommendations.

Reviewer #1:

Methods

Line 156. Please, consider changing the term White by European descents, as the latter term better describes the sample's ancestral origins.

Response: Thank you. We corrected the text of the Materials and Methods section according to this recommendation.

Reviewer #1:

Methods

Line 162. I suggest using PCs instead of principal components to be consistent with the abbreviation of PC previously used in this manuscript.

Response: Thank you for noticing this. We corrected the text, and now “PCs” is indicated instead of “principal components”.

Reviewer #1:

Methods

Did the authors evaluate whether the SNPs associated with the exposure were associated with the outcome?

Response: Thank you. Summary statistics for the association between IVs used in the main analysis and VVs (for Gene ATLAS, eMERGE, and FinnGen cohorts) is now provided in S5 Table (columns named “Association between IV and VVs”). Summary statistics for the association between IVs used in the sensitivity analysis and VVs is now provided in S3 Table (columns named “Varicose veins of lower extremities (OUTCOME)”).

All IVs were either not associated with VVs or the associations were not genome-wide significant.

Reviewer #1:

Results

Please, include the confidence intervals for all the reported estimates.

Response: Thank you. We included ORs and 95% CIs in the text of the Results section, in Table 1 and Table 2 (for the meta-analysis of MR results for FinnGen, eMERGE, and Gene ATLAS cohorts), and added all ORs and 95% CIs in Supplementary tables with MR results: S4 Table, S5 Table, S6 Table, and S7 Table.

Reviewer #1:

Results

Please report the F-statistic and include the proportion of variance explained (R2) of the IVs.

Response: Thank you. We added this information in S1 Table for the main analysis and S3 Table for the sensitivity analysis.

Reviewer #1:

Discussion

It would be interesting if the authors discuss if there is any potential confounders in the causal association between CD209 and VVs.

Response: Thank you. We included this information to the end of the Discussion section.

Hypothetically, potential confounders could be related to the inflammatory pathways which may promote the release of soluble form of CD209 and at the same time play a role in the promotion of VVs formation. However, as far as we are aware, to date, evidence from the literature does not allow to specify such common pathways

Reviewer #1:

Discussion

Please include as a potential limitation that analyzed data were obtained only from European descents and results could not be generalizable to other populations.

Response: Thank you. We added this limitation in the Discussion section.

---

## [Decision Letter · Decision Letter 1]

6 May 2022

Mendelian randomization analysis of plasma levels of CD209 and MICB proteins and the risk of varicose veins of lower extremities

PONE-D-22-02162R1

Dear Dr. Shadrina,

We’re pleased to inform you that your manuscript has been judged scientifically suitable for publication and will be formally accepted for publication once it meets all outstanding technical requirements.

Kind regards,

Renato Polimanti, Ph.D.

Academic Editor

PLOS ONE

Additional Editor Comments (optional):

Reviewers' comments:

Reviewer's Responses to Questions

**Comments to the Author**

1. If the authors have adequately addressed your comments raised in a previous round of review and you feel that this manuscript is now acceptable for publication, you may indicate that here to bypass the “Comments to the Author” section, enter your conflict of interest statement in the “Confidential to Editor” section, and submit your "Accept" recommendation.

Reviewer #1: All comments have been addressed

2. Is the manuscript technically sound, and do the data support the conclusions?

Reviewer #1: Yes

3. Has the statistical analysis been performed appropriately and rigorously? 

Reviewer #1: Yes

4. Have the authors made all data underlying the findings in their manuscript fully available?

Reviewer #1: Yes

5. Is the manuscript presented in an intelligible fashion and written in standard English?

Reviewer #1: Yes

6. Review Comments to the Author

Reviewer #1: I have no further comments to the authors. The authors have adequately addressed my comments in the revised version.

7. PLOS authors have the option to publish the peer review history of their article (what does this mean?). If published, this will include your full peer review and any attached files.

Reviewer #1: No

---

## [Editor Report · Acceptance letter]

13 May 2022

PONE-D-22-02162R1 

Mendelian randomization analysis of plasma levels of CD209 and MICB proteins and the risk of varicose veins of lower extremities 

Dear Dr. Shadrina:

I'm pleased to inform you that your manuscript has been deemed suitable for publication in PLOS ONE. Congratulations! Your manuscript is now with our production department. 

Kind regards, 

on behalf of

Dr. Renato Polimanti 

Academic Editor

PLOS ONE